# ATTENTIONNCE: CONTRASTIVE LEARNING WITH INSTANCE ATTENTION

## ABSTRACT

Contrastive learning has found extensive applications in computer vision, natural language processing, and information retrieval, significantly advancing the frontier of self-supervised learning. However, the limited availability of labels poses challenges in contrastive learning, as the positive and negative samples can be noisy, adversely affecting model training. To address this, we introduce instance-wise attention into the variational lower bound of contrastive loss, and proposing the AttentionNCE loss accordingly. AttentioNCE incorporates two key components that enhance contrastive learning performance: First, it replaces instance-level contrast with attention-based sample prototype contrast, helping to mitigate noise disturbances. Second, it introduces a flexible hard sample mining mechanism, guiding the model to focus on high-quality, informative samples. Theoretically, we demonstrate that optimizing AttentionNCE is equivalent to optimizing the variational lower bound of contrastive loss, offering a worst-case guarantee for maximum likelihood estimation under noisy conditions. Empirically, we apply AttentionNCE to popular contrastive learning frameworks and validate its effectiveness. The code is released at: `https://anonymous.4open.science/r/AttentioNCE-55EB`

## 1 INTRODUCTION

The pursuit of learning effective feature representations from unlabeled data has long been a long-standing goal in machine learning Wu et al. (2018); Zhuang et al. (2019); Chuang et al. (2020); Chu et al. (2023). Contrastive learning, as a powerful branch of self-supervised learning, driven by the pretext tasks of contrasting semantically similar positive examples with semantically unrelated negative examples to facilitate model pretraining. Contrastive learning and has demonstrated promising results Grill et al. (2020); Liu et al. (2021); Tong et al. (2023), garnering extensive adoption across various domains such as computer vision Chen et al. (2020a), natural language processing Radford et al. (2021); Luo et al. (2023), information retrieval Liu & Wang (2023), and other domains, even outperforms supervised learning in certain tasks Misra & Maaten (2020); He et al. (2020).

Within instance-level contrastive learning Wu et al. (2018); Oord et al. (2018); Chen et al. (2020a); Grill et al. (2020); He et al. (2020), positive and negative labels are typically assigned based on co-occurrence Liu et al. (2021) of input data, which often leads to label noise. For instance, in methods like CPC, SimCLR, and MOCO, positive examples are samples that co-occur with anchor data (e.g., augmented images or multimodal signals in videos). However, these positive examples may not always share semantic meaning with the anchor data, such as in cases of excessive cropping in images, leading to false positives. Negative instances typically comprise random samples that do not co-occur with the anchor data, yet they can unintentionally share semantic similarities with the anchor, resulting in false negatives. This noise arises from the absence of supervisory signals and the reliance on co-occurrence for automatic labeling, a process that inevitably generates both false positives and false negatives.

Noisy labels pose several challenges for contrastive learning. Firstly, they complicate likelihood modeling. The process of optimizing contrastive loss essentially aims to maximize the likelihood of identifying positive samples from a set of negatives Li et al. (2021). However, since the ground -truth labels for both positive and negative samples are not available, likelihood modeling becomes complicated. Secondly, the occurrence of false positives and false negatives result in semantically

unrelated samples are erroneously pulled together while semantically related samples are pushed apart, which will disrupt the semantic structure of embeddings Wang & Liu (2021). Previous research Chuang et al. (2022; 2020); Robinson et al. (2021); Wu et al. (2023) has shown that the presence of false positive and false negative examples significantly lead to performance drop. Moreover, while contrastive learning benefits from the mining of hard samples Robinson et al. (2021), existing research predominantly concentrates on hard negative mining, overlooking the potential benefits that could arise from the incorporation of hard positive mining. This oversight restricts the potential for further performance enhancements.

In this paper, we introduce a latent space to decompose the contrastive loss and derive its variational lower bound, which results in the proposal of the AttentionNCE loss as an alternative optimization target. AttentionNCE enables us to optimize the contrastive loss indirectly and provides a worst - case guarantee for maximum likelihood estimation under noisy conditions. To tackle the challenge of noisy perturbations, we utilize an attention mechanism to derive sample prototypes for contrast. By aggregating information from multiple samples, these prototypes assist in mitigating the effects of noisy perturbations during instance - level contrast. Finally, as we realize that high - quality hard samples can enhance contrastive learning, we incorporate a flexible and lightweight mechanism to mine both hard positive and hard negative samples, ensuring that the model focuses on high - quality and informative samples.

The contributions of this paper can be summarized as follows:

- We propose the AttentionNCE contrastive loss and theoretically prove that optimizing AttentionNCE is equivalent to optimizing the variational lower bound of the original contrastive loss, which provides a worst - case guarantee for maximum likelihood estimation (MLE) under noisy conditions.

- AttentionNCE incorporates attention - based sample prototype contrast to alleviate the impact of noise perturbations. Moreover, it includes a flexible hard - sample - mining mechanism to guide the model to focus on high - quality and informative samples.

- We apply the AttentionNCE loss to popular contrastive learning frameworks and validate its effectiveness.

## 2 RELATED WORK

**Self-supervised learning** is a branch of unsupervised learning that aims to exploit the internal structure of data Wu et al. (2018) for learning without relying on manual annotations. It achieves this through carefully designed pretext tasks He et al. (2020). These tasks typically include predicting arbitrary parts of the input based on observed parts Liu et al. (2021), such as autoencoder-based reconstruction Bengio et al. (2006), context prediction Doersch et al. (2015), colorization Zhang et al. (2016), rotation prediction Gidaris et al. (2018), among others. Another common type of pretext task involves predicting similar or not, or formulating it as classifying semantically related positive samples from semantically unrelated negative samples Gutmann & Hyvärinen (2010); Oord et al. (2018). This paradigm is also known as contrastive learning, which relieves the encoder from pixel-level information reconstruction and has shown promising results in various tasks Robinson et al. (2021); Tian et al. (2020); Wang & Isola (2020); Saunshi et al. (2019).

**Contrastive learning** has garnered significant attention in recent years as a self-supervised technique for representation learning Bachman et al. (2019); Hjelm et al. (2019); Henaff (2020); Misra & Maaten (2020); Wang & Isola (2020); Chen et al. (2020b); Radford et al. (2021); Li et al. (2021). Although the specific choices of representation encoder $f$ and similarity measure may vary depending on the task Gutmann & Hyvärinen (2010); Devlin et al. (2018); He et al. (2020); Dosovitskiy et al. (2014), they all share a common underlying principle of bringing positive pairs closer while pushing negative pairs apart to train the representation encoder $f$ through the optimization of a contrastive loss Wang & Isola (2020); Gutmann & Hyvärinen (2010); Oord et al. (2018); Hjelm et al. (2018); Wang & Isola (2020). Building upon this idea, several popular contrastive learning frameworks have been proposed, such as SimCLR Chen et al. (2020a), MOCO He et al. (2020), BYOL Grill et al. (2020), and SimSiam Chen & He (2021). However, in the self-supervised setting of contrastive learning, the issues of false positive samples Chuang et al. (2022) and false negative

examples Chuang et al. (2020); Robinson et al. (2021) can arise, which can degrade the performance of contrastive learning Wang & Liu (2021); Wu et al. (2023).

## 3 METHOD

### 3.1 CONTRASTIVE LEARNING AS MAXIMUM LIKELIHOOD ESTIMATION

Let $X$ denotes a set of samples $\{x^+, x_1^-, \cdots, x_N^-\}$, where $x^+$ denotes a positive sample that is semantically related to anchor $x$, while $x_1^-, \cdots, x_N^-$ represent negative samples that are semantically unrelated to the anchor $x$. We consider an embedding function $f_\theta$ parameterized by $\theta$, which maps a sample $x$ to a normalized $d$-dimensional embedding $f(x)$, let $\mathbf{q} = f(x)$ represent the embedding of an anchor (query) $x$. The embedding of a positive example is denoted as $\mathbf{k}^+ = f(x^+)$, while the embedding of a negative example is denoted as $\mathbf{k}^- = f(x^-)$.

The probability of classifying a positive sample from a set of N negative samples is modeled using following conventional parametric softmax formulation Oord et al. (2018):

$$P(X|\theta) = \frac{\exp(\mathbf{q}^\mathsf{T}\mathbf{k}^+/\tau)}{\exp(\mathbf{q}^\mathsf{T}\mathbf{k}^+/\tau) + \sum_{j=1}^{N} \exp(\mathbf{q}^\mathsf{T}\mathbf{k}_j^-/\tau)}, \tag{1}$$

where $\mathbf{q}^T\mathbf{k}$ measures the similarity between the query and key, while $\tau$ is a temperature scalling that controls the concentration of the softmax distribution. Equation 1 describes the likelihood of classifying the positive key from $N$ negative keys, parameterized by the weights of embedding function $f_\theta$. The maximum likelihood is achieved when the embedding of the positive pair, i.e., the similarity between the query and the positive key $\mathbf{q}^\mathsf{T}\mathbf{k}^+ \to +\infty$, or when the embedding of the negative pair, i.e., the similarity between the query and the negative key $\mathbf{q}^\mathsf{T}\mathbf{k}_j^- \to -\infty, j \in \{1, 2, \cdots N\}$. It is important to note that the embedding induces a metric over the sample space $d(x, x^+) = ||\mathbf{q} - \mathbf{k}^+||$. For embeddings lies on the surface of a hypersphere of radius $1/\tau$, there exists a one-to-one correspondence between Euclidean distance and similarity $d(x, x^+) = \sqrt{2/\tau^2 - 2\mathbf{q}^\mathsf{T}\mathbf{k}^+}$. As a result, the maximum likelihood estimation process serves as a means to effectively bring positive samples closer to the anchor and push negative samples further apart. Therefore, the process of maximum likelihood of classifying the positive key from $N$ negative keys in equation 1 is also the process of finding the optimal parameter $\theta$ that maps semantically related positive samples to be close in distance, while ensuring that semantically unrelated negative sample pairs are mapped to be far apart.

It is worth noting that there is a discrepancy in the interpretation of equation 1 in the literature. Li et al. (2021) interprets equation 1 as likelihood, while Oord et al. (2018) interprets it as a posterior. The main reason for this difference lies in the different interpretations of the semantics of matching scores. However, we do not distinguish between the concepts of maximum likelihood or maximum posteriori arising from the semantics of positive example scores. This conceptual difference does not affect the subsequent methods and theories presented in this paper.

In practice, it is common to maximize the logarithm of the equation above, which yields the popular InfoNCE loss

$$\mathcal{L}_{\text{InfoNCE}} = -\mathbb{E} \, \log \frac{\exp(\mathbf{q}^\mathsf{T}\mathbf{k}^+/\tau)}{\exp(\mathbf{q}^\mathsf{T}\mathbf{k}^+/\tau) + \sum_{j=1}^{N} \exp(\mathbf{q}^\mathsf{T}\mathbf{k}_j^-/\tau)}. \tag{2}$$

### 3.2 DECOMPOSITION OF THE CONTRASTIVE LOSS

In the presence of noise, our set of examples $\{x^+, x_1^-, \ldots, x_N^-\}$ contains false positives and false negatives, leading to unreliable positive and negative keys. This complicates the maximum likelihood estimation outlined in equation 2. To tackle this challenge, we begin by decomposing the contrastive loss. We define $\mathbf{h} = [\mathbf{h}^{\text{pos}}, \mathbf{h}_1^{\text{neg}}, \ldots, \mathbf{h}_N^{\text{neg}}]$, representing the prototype features for both positive and negative samples. The task of identifying a positive sample among negative ones is linked to the latent variable, which we can model similarly to equation 1 using a softmax formulation. By introducing a distribution $q(\mathbf{h})$ over $\mathbf{h}$, we can further decompose the contrastive loss as

follows:

$$
\begin{aligned}
\log P(X|\theta) &= \sum_{\mathbf{h}} q(\mathbf{h}) \log P(X|\theta) = \sum_{\mathbf{h}} q(\mathbf{h}) \log \frac{P(X,\mathbf{h}|\theta)}{P(\mathbf{h}|X,\theta)} \\
&= \sum_{\mathbf{h}} q(\mathbf{h}) \log(\frac{P(X,\mathbf{h}|\theta)}{q(\mathbf{h})} \frac{q(\mathbf{h})}{P(\mathbf{h}|X,\theta)}) \\
&= \sum_{\mathbf{h}} q(\mathbf{h}) \log(\frac{P(X,\mathbf{h}|\theta)}{q(\mathbf{h})}) d\mathbf{h} + \sum_{\mathbf{h}} q(\mathbf{h}) \log \frac{q(\mathbf{h})}{P(\mathbf{h}|X,\theta)} \\
&= \sum_{\mathbf{h}} q(\mathbf{h}) \log \frac{P(X,\mathbf{h}|\theta)}{q(\mathbf{h})} + \mathbb{KL}(q(\mathbf{h})||P(\mathbf{h}|X,\theta)).
\end{aligned}
\tag{3}
$$

$q(\mathbf{h})$ denotes some distribution over $\mathbf{h}$, and $\sum_{\mathbf{h}} q(\mathbf{h}) = 1$. The first term $\mathcal{J}(\theta) = \sum_{\mathbf{h}} q(\mathbf{h}) \log \frac{P(X,\mathbf{h}|\theta)}{q(\mathbf{h})}$ in the above equation is the variational lower bound Kingma & Welling (2013). We can rewrite the likelihood function as:

$$
\log P(X|\theta) = \mathcal{J}(\theta) + \mathbb{KL}(q(\mathbf{h})||P(\mathbf{h}|X,\theta)).
\tag{4}
$$

Since the KL divergence is non-negative, $\mathcal{J}(\theta)$ lower bounds the likelihood, that is,

$$
\log P(X|\theta) \geq \mathcal{J}(\theta).
\tag{5}
$$

In the presence of noise, maximum likelihood estimation is infeasible. However, the variational lower bound $\mathcal{J}(\theta)$ providing a worst-case guarantee for maximizing the log-likelihood $\log P(X|\theta)$.

### 3.3 SAMPLE PROTOTYPE WITH INSTANCE ATTENTION

Note that the variational lower bound $\mathcal{J}(\theta)$ incorporates a latent variable, $\mathbf{h} = [\mathbf{h}^{\mathrm{pos}}, \mathbf{h}_1^{\mathrm{neg}}, \cdots, \mathbf{h}_N^{\mathrm{neg}}]$, which we instantiate as a sample prototype derived from the attention mechanism. The motivation behind this design is straightforward. First, the attention mechanism guides the model to focus on high-quality samples, ensuring that the sample prototype captures richer and more relevant features. Second, by generating $\mathbf{h}$ as a prototype from multiple sample features, rather than performing instance-level contrasts, it becomes more robust to noise in contrastive learning.

**Attention for positive samples**: We first obtain $M$ views of a sample. For instance, we apply random augmentations to the anchor sample $x$, resulting in a set of $M$ positive samples (views) with their corresponding positive keys $\{\mathbf{k}_i^+\}_{i=1}^M$:

$$
\mathbf{k}_i^+ = f_\theta(\mathcal{T}(x)), i \in \{1, 2, \cdots M\}.
\tag{6}
$$

$\mathcal{T}(\cdot)$ is a family of random data augmentations, $f_\theta(\cdot)$ is the embedding function. Among these $M$ positive keys, in order to direct the model's attention towards positive keys that encodes more class information, and ignore the keys of semantically irrelevant false-positive examples, we introduce attention for the $M$ positive keys, which maps a query $\mathbf{q}$ and a set of positive keys $\{\mathbf{k}_i^+\}_{i=1}^M$ to a positive key

$$
\mathbf{h}^{\mathrm{pos}} = \sum_{i=1}^M \alpha_i \mathbf{k}_i^+,
\tag{7}
$$

where $\alpha_i$ represents the weight assigned to each positive key, which is computed based on the similarity between the query and the corresponding key

$$
\alpha = \mathrm{softmax}(\mathbf{q}^\mathsf{T}\mathbf{k}_1^+/d_{\mathrm{pos}}, \mathbf{q}^\mathsf{T}\mathbf{k}_2^+/d_{\mathrm{pos}}, \cdots, \mathbf{q}^\mathsf{T}\mathbf{k}_M^+/d_{\mathrm{pos}}).
\tag{8}
$$

$d_{\mathrm{pos}}$ is scaling factor that controls the concentration of attention on the most reliable positive samples with the highest similarity. Applying the attention function to the features from $M$ views, the resultant positive feature can be interpreted as a prototype of positive instances, encoding a richer set of class-specific information.

**Attention for negative samples**: For negative examples, the same attention mechanism is applied to the negative example keys $\mathbf{k}_j^-$ as well

$$\mathbf{h}_j^{\text{neg}} = \beta_j \mathbf{k}_j^-, \tag{9}$$

where

$$\beta = \text{softmax}(\mathbf{q}^\mathsf{T}\mathbf{k}_1^-/d_{\text{neg}}, \mathbf{q}^\mathsf{T}\mathbf{k}_2^-/d_{\text{neg}}, \cdots, \mathbf{q}^\mathsf{T}\mathbf{k}_M^-/d_{\text{neg}}) \cdot N. \tag{10}$$

$d_{\text{neg}}$ is scaling factor that controls the concentration of attention on the most hard samples with the highest similarity. The scalar multiplier $N$ in the equation 10 is designed to ensure that the sum of weights for the $N$ negative samples is equal to $N$, aligning the negative sample weight sum in the InfoNCE loss to $N$ (where each sample has a weight of 1). This prevents the computation of an excessively large or small loss value.

**Hard sample mining effect**: For a given anchor point, when the similarity scores of other samples relative to the anchor point are sorted in ascending order, the negative samples with relatively high similarity and the positive samples with relatively low similarity are closer to the decision boundary, as shown in the white area in Fig. 1. These samples are also known as hard samples. Focusing on such samples is beneficial for representation learning Robinson et al. (2021) since it

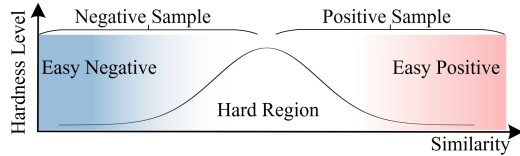

Figure 1: Hard sample mining effect.

helps the model learn more accurate decision boundaries Liu & Wang (2023). Attention-based sample prototypes inherently include a flexible hard sample mining mechanism. This mechanism enables the prototype to capture a richer features of hard samples near the decision boundary.

Specifically, during the generation of positive prototypes, a larger $d_{pos}$ value can result in more hard positive samples with low similarity being incorporated into the positive prototypes; while during the generation of negative prototypes, a smaller $d_{neg}$ value can lead to more hard negative samples with high similarity being included in the negative prototypes. Consequently, the hard sample mining effect of AttentionNCE can be flexibly achieved by setting the scaling factors $d_{pos}$ and $d_{neg}$.

### 3.4 DEVIRIATION OF ATTENTIONNCE

After deriving the instance prototypes from the attention mechanism, this section will integrate these prototypes into the variational lower bound, transforming $\mathcal{J}(\theta)$ into a practical alternative optimization target. We first simplify the optimization objective $\mathcal{J}(\theta)$:

$$\begin{aligned} \arg\max_\theta \mathcal{J}(\theta) &= \arg\max_\theta \sum_{\mathbf{h}} q(\mathbf{h}) \log P(X, \mathbf{h}|\theta) - \sum_{\mathbf{h}} q(\mathbf{h}) \log q(\mathbf{h}) \\ &= \arg\max_\theta \sum_{\mathbf{h}} q(\mathbf{h}) \log P(X, \mathbf{h}|\theta). \end{aligned} \tag{11}$$

Given the freedom to choose any distribution for $q(\mathbf{h})$, based on equation 7 and equation 9, the distribution $q(\mathbf{h})$ is selected as follows:

$$q(\mathbf{h}) = \begin{cases} 1, & \text{if } \mathbf{h} = [\mathbf{h}^{\text{pos}}, \mathbf{h}_1^{\text{neg}}, \cdots, \mathbf{h}_N^{\text{neg}}] \\ 0, & \text{otherwise.} \end{cases} \tag{12}$$

So

$$\begin{aligned} \arg\max_\theta \mathcal{J}(\theta) &= \arg\max_\theta \sum_{\mathbf{h}} \mathbb{1}(\mathbf{h} = [\mathbf{h}^{\text{pos}}, \mathbf{h}_1^{\text{neg}}, \cdots, \mathbf{h}_N^{\text{neg}}]) \cdot \log P(X, \mathbf{h}|\theta) \\ &= \arg\max_\theta \log P(X, \mathbf{h}|\theta) = \arg\max_\theta \log P(X|\mathbf{h}, \theta) + \log P(\mathbf{h}|\theta). \end{aligned}$$

Assuming a uniform prior distribution, $P(\mathbf{h}|\theta)$, for the latent variables, and model the likelihood $P(X|\mathbf{h}, \theta)$ similar to equation 1 using softmax formulation, we have

$$\arg\max_\theta \mathcal{J}(\theta) = \arg\max_\theta \log \frac{\exp(\mathbf{q}^\mathsf{T}\mathbf{h}^{\text{pos}}/\tau)}{\exp(\mathbf{q}^\mathsf{T}\mathbf{h}^{\text{pos}}/\tau) + \sum_{j=1}^N \exp(\mathbf{q}^\mathsf{T}\mathbf{h}_j^{\text{neg}}/\tau)} + const. \tag{13}$$

The above equation presents an equivalent optimization objective for the variational lower bound, utilizing sample prototypes obtained through the attention mechanism as updated keys. This allows the attention mechanism to be fully integrated into the variational lower bound, resulting in the novel **Attention** based Info**NCE** (AttentionNCE)

$$\mathcal{L}_{\text{AttentionNCE}} = -\mathbb{E} \log \frac{\exp(\mathbf{q}^{\mathsf{T}} \mathbf{h}^{\text{pos}}/\tau)}{\exp(\mathbf{q}^{\mathsf{T}} \mathbf{h}^{\text{pos}}/\tau) + \sum_{j=1}^{N} \exp(\mathbf{q}^{\mathsf{T}} \mathbf{h}_{j}^{\text{neg}}/\tau)}. \tag{14}$$

The expectation is taken over the tuple of $(x, x_1^+, \cdots, x_M^+, x_1^-, \cdots, x_N^-)$, which means that the computation of each loss value requires an anchor point, $M$ positives, and $N$ negatives.

Equation 13 presents the first theoretical insight: optimizing the AttentionNCE loss is equivalent to optimizing the variational lower bound on the contrastive loss. The second theoretical finding, shown in equation 4 is that the gap between the ideal contrastive loss and the variational lower bound is governed by the KL divergence. Due to the non-negativity of KL divergence, the AttentionNCE loss always lower bounds the InfoNCE loss, as illustrated in Fig. 2. Therefore, by optimizing the AttentionNCE loss, we indirectly optimize the contrastive loss, establishing a theoretical foundation for AttentionNCE as an effective alternative optimization objective. This means that AttentionNCE offers a

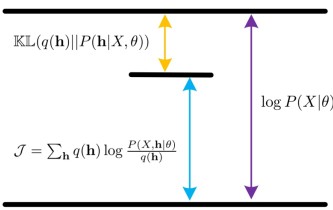

Figure 2: The relationship between InfoNCE and AttentionNCE loss.

worst-case guarantee for maximum likelihood estimation under noisy conditions. Furthermore, by leveraging sample prototypes obtained through the attention mechanism for contrast, AttentionNCE not only reduces impact of individual noise samples but also directs the model's focus toward higher-quality samples.

It is also important to note that when $q(h) = P(\mathbf{h}|X, \theta)$, that is $\mathbb{KL}(q(\mathbf{h})||P(\mathbf{h}|X, \theta)) = 0$, indicating that AttentionNCE loss provides a tighter lower bound. If we assume that the distribution chosen in equation 12 represents the true posterior distribution of sample prototypes, then AttentionNCE can be formalized as an Expectation-Maximization (EM) algorithm Dempster et al. (1977), where the attention function in the equation 7 and equation 9 corresponds to the expectation step, and maximizing the AttentionNCE loss corresponds to the maximization step.

### 3.5 IMPLEMENTATION OF ATTENTIONNCE.

The implementation of AttentionNCE is straightforward and can be summarized in three steps as dipicted in Fig 3: Step 1, encode M positive examples to obtain their feature representations as positive keys, and encode N negative examples to obtain their feature representations as negative keys. Step 2, using the feature representation of the anchor point as the query, apply an attention function to the query and all positive and negative keys, resulting in updated positive and negative keys. Step 3, compute the standard contrastive loss based on the updated positive and negative keys. The pseudocode for the AttentionNCE loss is presented in Algorithm 1.

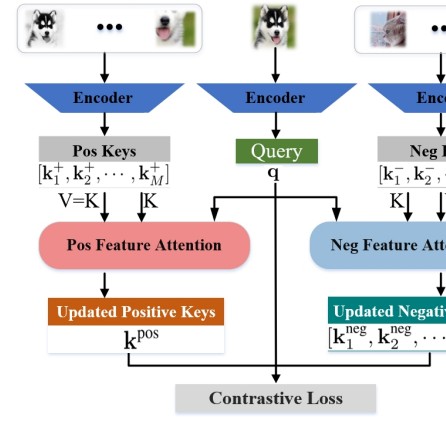

Figure 3: Flowchart of AttentionNCE.

**Complexity**: For the standard contrastive loss, the matching scores $\mathbf{q}^{\mathsf{T}}\mathbf{k}$ between the anchor point and all keys need to be computed. Therefore, compared to the standard contrastive loss, AttentionNCE introduces additional computational overhead primarily in two aspects: (i) Encoding $M-1$ additional positive examples in line 1. In contrast to standard contrastive learning, which encodes only one anchor point, one positive example, and $N$ negative examples (with a time complexity of

---

**Algorithm 1:** Pseudocode for AttentionNCE.

---

**Input:** Anchor $x$, $M$ positive samples $\{x_i^+\}_{i=1}^M$, $N$ negative samples $\{x_j^-\}_{j=1}^N$, encoder $f_\theta(\cdot)$,
scalling factor $d_{\text{pos}}$, $d_{\text{neg}}$.
**Output:** AttentionNCE loss.

1 $\mathbf{q} = f_\theta(x)$, $\{\mathbf{k}_i^+\}_{i=1}^M = \{f_\theta(x_i^+)\}_{i=1}^M$, $\{\mathbf{k}_j^-\}_{j=1}^N = \{f_\theta(x_j^-)\}_{j=1}^N$ ;
2 Update positive key via equation 7 ;
3 Update negative keys via equation 9;
4 Calculate AttentionNCE via equation 14;

**Result:** AttentionNCE loss.

---

$\mathcal{O}(2+N)$), AttentionNCE additionally encodes $M-1$ positive samples, resulting in a time complexity of $\mathcal{O}(1+M+N)$. However, since $M$ is typically a small constant such as 4, the time complexity remains linear compared to standard contrastive learning. (ii) Applying positive feature attention and negative feature attention in line 2 and 3. This computational complexity can be considered negligible compared to the encoding of a single example. This is because we can leverage the additivity property of inner product operations, where $\mathbf{q}^\mathsf{T}\mathbf{h}^{\text{pos}} = \mathbf{q}^\mathsf{T}\sum_{i=1}^M \alpha_i \mathbf{k}_i^+ = \sum_{i=1}^M \alpha_i \mathbf{q}^\mathsf{T}\mathbf{k}_i^+$, and $\mathbf{q}^\mathsf{T}\mathbf{h}^{\text{neg}} = \mathbf{q}^\mathsf{T}\beta_j\mathbf{k}_j^- = \beta_j\mathbf{q}^\mathsf{T}\mathbf{k}_j^-$. So AttentionNCE only requires computing the matching scores once similar to the standard contrastive loss, and the actual additional cost in line 2 and 3 comes from calculating the attention weights in equation 8 and equation 10, which can be negligible.

**Relations to previous research**: Both AttentionNCE and CMC Tian et al. (2020) introduce multiple views. However, in CMC, one sample is fixed as the anchor point, and positives and negatives are enumerated from the other view to compute the standard contrastive loss. In contrast, we generate sample prototypes through multiple views and optimize the variational lower bound. ProtoNCE Li et al. (2021) also considers prototype contrast, yet the prototype in ProtoNCE represents the class center in unsupervised clustering, which differs from the sample prototypes generated by the attention function in this work. Additionally, HCL Robinson et al. (2021) also takes into account the mining of hard negative samples. However, HCL achieves this by assuming VMF distribution for negative keys, while our approach mines hard negative samples via the attention mechanism. Moreover, only AttentionNCE encompasses the mining of hard positive samples.

## 4 EXPERIMENTS

### 4.1 PERFORMANCE ON SMALL-SCALE DATASETS

SimCLR Chen et al. (2020a) applies two rounds of augmentation to $N$ samples, resulting in $2N$ samples. The two views of the same sample are positive pairs, while the remaining $2N-2$ samples serve as negative examples. The standard InfoNCE loss, also referred to as NT-Xent loss in the original paper, is then computed. To ensure a fair comparison, we follow the same settings as Chen et al. (2020a); Chuang et al. (2020); Robinson et al. (2021), including ResNet50 He et al. (2016) architecture, data augmentation methods, learning rate, Adam optimizer Kingma & Ba (2014), and adhering to the same linear evaluation protocol. The only modification is replacing the InfoNCE loss with the AttentionNCE loss to pretrain the model. Detailed settings are presented in Tabel 5 in Appendix. Table 1 presents the top-1 accuracy evaluation results on CIFAR-10/100 Krizhevsky & Hinton (2009), STL-10 Coates et al. (2011), and TinyImageNet Le & Yang (2015), with the optimal and second-best results marked in bold and underlined, respectively.

The results for the comparative methods in lines 1-3 are reported from Robinson et al. (2021), lines 5-6 from HaoChen et al. (2021), lines 7-9 from Wang et al. (2021), and lines 10-11 from Zhang et al. (2022). While using the same experimental settings as prior work Chen et al. (2020a); Chuang et al. (2020); Robinson et al. (2021), AttentionNCE demonstrates significant improvements over the baseline method SimCLR that uses InfoNCE loss. Notably, even within 200 training epochs, AttentionNCE surpasses the performance SimCLR achieves in 400 epochs. Moreover, Figure 4 provides a t-SNE visualization of instance features on CIFAR-10, illustrating that the AttentionNCE loss enables earlier and clearer separation between classes compared to InfoNCE.

Table 1: Linear Evaluation on Small-scale Datasets.

| Line | Method | Encoder | CIFAR10 | | STL10 | | CIFAR100 | | Tiny-ImageNet | |
|------|--------|---------|---------|---------|---------|---------|---------|---------|---------|---------|
| | | | ep200 | ep400 | ep200 | ep400 | ep200 | ep400 | ep200 | ep400 |
| 1 | SimCLR (Chen et al. (2020a)) | ResNet50 | 89.2 | 91.1 | 78.5 | 80.2 | 64.0 | 66.4 | 51.6 | 53.4 |
| 2 | DCL (Chuang et al. (2020)) | ResNet50 | _91.7_ | _92.1_ | 81.6 | 84.3 | 65.5 | 67.7 | 52.2 | 53.7 |
| 3 | HCL (Robinson et al. (2021)) | ResNet50 | 91.5 | 91.9 | _85.5_ | 87.2 | 66.3 | _69.5_ | _55.4_ | _57.0_ |
| 4 | RINCE (Chuang et al. (2022)) | ResNet50 | - | 91.6 | - | - | - | - | - | - |
| 5 | SimSam (Chen & He (2021)) | ResNet50 | 87.5 | 90.3 | - | - | 61.6 | 65.0 | 34.8 | 39.5 |
| 6 | SpectralCL (HaoChen et al. (2021)) | ResNet50 | 88.7 | 90.2 | - | - | 62.5 | 65.8 | 41.3 | 45.4 |
| 7 | NPID (Wu et al. (2018)) | ResNet50 | - | 79.1 | - | 80.8 | - | 51.6 | - | - |
| 8 | NPID+CLD (Wang et al. (2021)) | ResNet50 | - | 86.7 | - | 83.6 | - | 57.5 | - | |
| 9 | MOCO+CLD (Wang et al. (2021)) | ResNet50 | - | 87.5 | - | 84.3 | - | 58.1 | - | |
| 10 | SimMoCo (Zhang et al. (2022)) | ResNet18 | 82.4 | - | 80.6 | - | 54.1 | - | - | - |
| 11 | SimCo (Zhang et al. (2022)) | ResNet18 | 85.6 | - | 83.2 | - | 58.4 | - | - | - |
| 12 | SDMP (Ren et al. (2022)) | ResNet50 | 89.5 | - | - | - | 68.2 | - | - | - |
| 13 | $\alpha$-CL-direct (Tian (2022)) | ResNet50 | 90.1 | 91.2 | 84.7 | 87.9 | 66.3 | 68.5 | - | - |
| 14 | ADNCE (Wu et al. (2023)) | ResNet50 | 90.7 | 91.9 | 85.1 | _88.0_ | _66.9_ | 69.3 | - | - |
| 15 | AttentionNCE | ResNet50 | **92.4** | **93.1** | **87.1** | **89.4** | **69.8** | **70.2** | **56.6** | **58.6** |
| 16 | _vs. SimCLR_ | - | 3.2 ↑ | 2.0 ↑ | 8.6 ↑ | 7.2 ↑ | 5.8 ↑ | 3.8 ↑ | 5.0 ↑ | 5.2 ↑ |

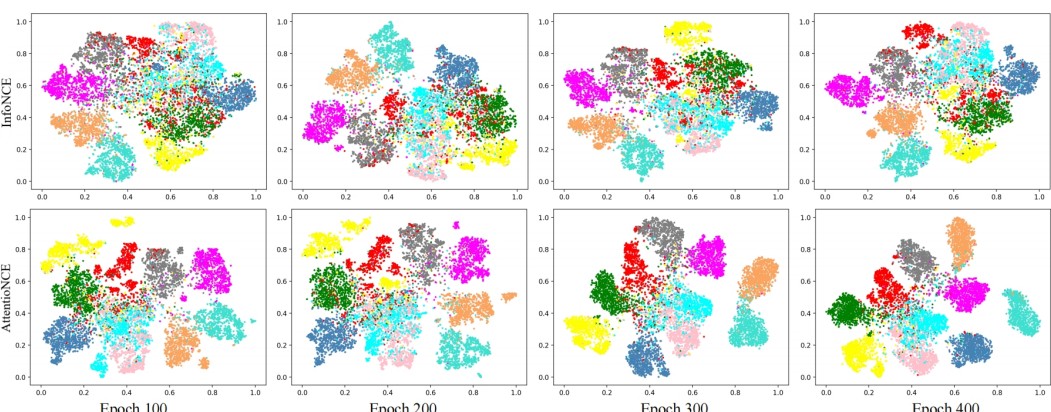

Figure 4: AttentionNCE loss has earlier and better separation between classes (indicated by the dot color) than InfoNCE loss in the t-SNE visualization of instance feature on CIFAR10.

## 4.2 PERFORMANCE ON IMAGENET

We evaluated our method on the widely used ImageNet benchmark. Specifically, we tested AttentionNCE within both the SimCLR and MoCo-v3 frameworks, keeping the training protocols and hyperparameter settings identical to those of SimCLR and MoCo-v3. The only modification was replacing the InfoNCE loss with our AttentionNCE loss ($d_{pos} = 4$, $d_{neg} = 1$, $M = 4$). Table 2 demonstrates significant performance improvements of AttentionNCE over InfoNCE in both Sim-CLR and MoCo-v3 frameworks. We also compare our results with state-of-the-art (SOTA) baselines, which enhance SimCLR through techniques such as dynamic dictionaries with momentum encoders (MoCo-v1/v2/v3), removing negative samples and using stop-gradient techniques (Sim-Siam, BYOL), or online clustering (SwAV). All the results of comparative methods are reported from Chuang et al. (2022). While our method may not outperform state-of-the-art approaches, AttentionNCE is orthogonal to these advancements. This means it can be seamlessly integrated with these state-of-the-art techniques to potentially further boost their performance, making Attention-NCE a complementary and promising enhancement in the field of contrastive learning.

## 4.3 FURTHER ANALYSIS

### 4.3.1 HOW DOES THE SCALING FACTOR AFFECT THE PERFORMANCE?

**Ablation Study on Hard Sample Mining**. Figure 5 shows how different combinations of $d_{pos}$ and $d_{neg}$ affect model performance on the CIFAR-10 and CIFAR-100 datasets. When both $d_{pos}$ and $d_{neg}$ are set to 10, the attention weight assigned to each sample approaches 1, effectively removing

Table 2: Linear Evaluation on ImageNet.

| Method | Backbone | Parameters | Improvement to SimCLR | Top-1 | Top-5 |
|---|---|---|---|---|---|
| Supervised He et al. (2016) | ResNet-50 | 24M | - | 76.5 | - |
| SimSiam Chen & He (2021) | ResNet-50 | 24M | No negative pairs | 71.3 | - |
| BYOL Grill et al. (2020) | ResNet-50 | 24M | No negative pairs | 74.3 | 91.6 |
| Barlow Twins Zbontar et al. (2021) | ResNet-50 | 24M | Redundancy reduction | 73.2 | 91.0 |
| SwAV Caron et al. (2020) | ResNet-50 | 24M | Cluster discrimination | 75.3 | - |
| SimCLR Chen et al. (2020a) | ResNet-50 | 24M | None | 69.3 | 89.0 |
| +RINCE Chuang et al. (2022) | ResNet-50 | 24M | Symmetry controller q | 70.0 | 89.8 |
| +AttentionNCE(Ours) | ResNet-50 | 24M | Attenition Contrast | 70.8 | 91.1 |
| MoCo He et al. (2020) | ResNet-50 | 24M | Momentum encoder | 60.6 | - |
| MoCo-v2 Chen et al. (2020c) | ResNet-50 | 24M | Momentum encoder | 71.1 | 90.1 |
| MoCo-v3 Chen et al. (2021) | ResNet-50 | 24M | Momentum encoder | 73.8 | - |
| +RINCE Chuang et al. (2022) | ResNet-50 | 24M | Symmetry controller q | 74.2 | 91.8 |
| +AttentionNCE(Ours) | ResNet-50 | 24M | Attenition Contrast | 74.6 | 91.9 |

the effect of hard sample mining. Under these conditions, the model achieves suboptimal results on both datasets, demonstrating that removing the hard sample mining mechanism has a negative impact on performance. At this point, compared to InfoNCE, AttentionNCE improves performance by 0.9% on CIFAR-10 and 2.4% on CIFAR-100. This improvement is primarily due to the use of sample prototypes, which help mitigate the noise compared to instance-level contrast. Next, we observe that larger values of $d_{pos}$ yield better results on both datasets, as higher attention is given to hard positive samples. This indicates that mining hard positive samples significantly boosts model performance. Finally, the selection of $d_{neg}$ differs between the two datasets: CIFAR-10 performs better with larger $d_{neg}$ values, whereas CIFAR-100 favors smaller $d_{neg}$ values. This suggests that hard negative sample mining plays a more critical role in CIFAR-100 than in CIFAR-10.

These differences can be attributed to the variation in negative sample noise rates between the two datasets. While CIFAR-10 and CIFAR-100 employ the same data augmentation strategy and therefore have identical positive noise rates, CIFAR-10, with only 10 classes, has a higher negative noise rate (1/10) compared to CIFAR-100 (1/100). Consequently, the model trained on CIFAR-10 is more prone to overfitting noisy samples, particularly during extended training. The memorization effect in deep neural networks Arpit et al. (2017) provides insight into this behavior. Specifically, deep models initially memorize clean training samples and gradually fit noisy data as training epochs increase Zhang et al. (2021); Han et al. (2018). Thus, while larger $d_{pos}/d_{neg}$ ratios enhance the model's ability to mine hard samples, they also increase the risk of overfitting noisy data. AttentionNCE provides a flexible approach for hard sample mining, but it requires balancing the exploration of hard (noisy) samples with the exploitation of easy (clean) samples.

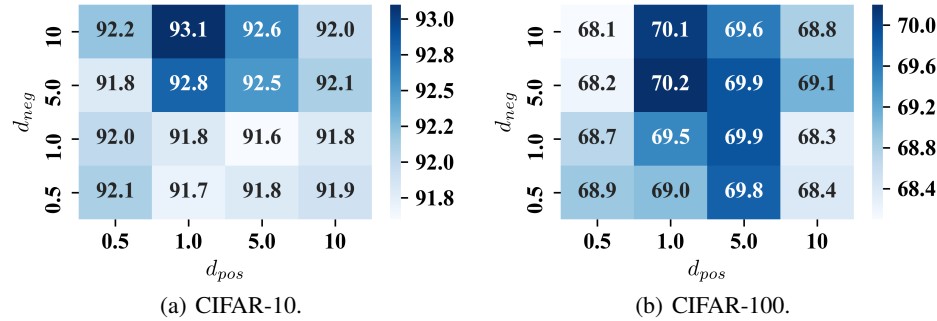

(a) CIFAR-10.            (b) CIFAR-100.

Figure 5: The impact of different $(d_{pos}, d_{neg})$ combinations on performance.

### 4.3.2 DOES A LARGER VALUE OF $M$ LEAD TO BETTER PERFORMANCE?

**Ablation Study on Sample Prototypes**. $M$ denotes the number of positive examples for generating positive sample prototype, with $M \geq 1$. Table 3 presents the effect of varying $M$ on top-1 linear

evaluation accuracy. When $M = 1$, meaning only one positive example is used to generate the positive prototype, the feature of the positive prototype is equivalent to the positive feature itself, effectively removing the influence of the attention based sample prototype. In this case, Attention-NCE produces suboptimal results on the CIFAR-10 and STL-10 datasets, indicating that removing attention-based sample prototypes negatively impacts performance. Secondly, increasing $M$ consistently improves performance. However, beyond $M = 3$, the benefits diminish, as four views already provide sufficient information for a given sample. At this point, the sample features derived from positive attention are representative enough. Therefore, setting $M$ too large is unnecessary and could introduce excessive computational overhead without yielding additional gains.

Table 3: Impacts of different $M$ on performance.

| Dataset | Encoder | SimCLR | AttentionNCE | | | | | |
|---|---|---|---|---|---|---|---|---|
| | | | M=1 | M=2 | M=3 | M=4 | M=5 | M=6 |
| CIFAR10 | ResNet50 | 91.1 | 92.2 | 92.5 | 92.8 | **93.1** | 93.0 | 93.2 |
| STL10 | ResNet50 | 80.2 | 85.4 | 86.5 | 87.8 | 89.4 | **89.5** | 89.4 |

## 4.4 ATTENTIONNCE IN SUPERVISED LEARNING

We show the performance of AttentionNCE in a supervised setting to further validate the motivation for using attention-based sample prototypes for contrast. Specifically, following the SimCLR framework, we apply augmentation to a batch of $N$ samples, which results in $2N$ samples. For any given sample (anchor point), positive samples are the remaining $2N - 1$ samples sharing the same label as the anchor point, while negative samples are those with different labels. This process eliminates the cost of false - negative samples. Subsequently, we calculate the AttentionNCE loss. For SimCLR, labels are utilized to exclude false - negative samples. The data augmentation methods, network architecture, optimizer, and linear evaluation protocol remain unchanged. Table 4 displays the performance of AttentionNCE in a supervised setting. Without the cost related to false negative samples, AttentionNCE shows a significant improvement. This is because the sample prototypes generated from multiple samples by means of attention can encode more information of semantic classes. In contrast to instance-level contrast, when using prototypes, they are capable of capturing more general and representative features within a semantic class, which is beneficial for reducing the influence of noise and variability within individual samples. This advantage backs up the underlying motivation for generating prototypes using the attention function for contrast.

Table 4: AttentionNCE under supervised settings.

| Method | CIFAR10 | | CIFAR100 | | Tiny-ImageNet | |
|---|---|---|---|---|---|---|
| | ep200 | ep400 | ep200 | ep400 | ep200 | ep400 |
| SimCLR(*Supervised*) | 93.1 | 93.6 | 64.6 | 68.6 | 52.7 | 54.4 |
| AttentionNCE(*Supervised*) | 93.9 | 94.2 | 73.3 | 73.4 | 58.9 | 60.2 |
| *vs. baseline* | 0.8 ↑ | 0.6 ↑ | 8.7 ↑ | 4.8 ↑ | 6.2 ↑ | 5.8 ↑ |

## 5 CONCLUSION AND LIMITATIONS

This paper introduces instance-level attention into contrastive learning by integrating attention-based sample prototypes into the variational lower bound of contrastive loss, resulting in the proposed AttentionNCE loss. AttentionNCE directs the model's focus toward more informative and relevant samples, offering a worst-case guarantee for maximum likelihood estimation under noisy conditions. Despite its simplicity and ease of implementation, AttentionNCE includes two key components that enhance performance. First, attention-based sample prototypes help mitigate the impact of noise in instance-level contrast. Second, the flexible incorporation of hard positive and hard negative sample mining further boosts performance. However, the balance between exploiting easy (clean) samples and exploring hard (potentially noisy) samples requires further study. This balance is crucial for generating more effective sample prototypes, $\mathbf{h}$, and achieving a tighter variational lower bound. We hope this study will inspire further theoretical analyses in self-supervised contrastive learning and promote the extension of instance-level attention mechanisms in future methods.

## ETHICS STATEMENT

The research presented in this paper fully adheres to the ICLR Code of Ethics.

## REPRODUCIBILITY STATEMENT

Implementation details are provided in Section 4.1. Additionally, we have released our code, datasets, and pre-trained models in the following repository: `https://anonymous.4open.science/r/AttentioNCE-55EB`.

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

## A  APPENDIX

In addition to the unique parameters of AttentionNCE, namely positive sample size $M$, positive scaling $d_{\text{pos}}$ and negative scaling $d_{\text{neg}}$, all other hyperparameters, data augmentation methods remain exactly the same as Chuang et al. (2020); Robinson et al. (2021). The detailed parameter settings for the main results are shown in Table 5. The results for the CIFAR10, CIFAR100, and STL10 datasets were obtained by running the experiments on a cloud server equipped with two NVIDIA GeForce RTX 3090 GPUs. The results for the Tiny-ImageNet dataset were obtained on a cloud server with one NVIDIA A100 40GB GPU. All the code and pre-trained models have been released at: `https://anonymous.4open.science/r/AttentioNCE-55EB`.

Table 5: Detailed parameter settings.

| Parameter Settings | CIFAR10 | STL10 | CIFAR100 | Tiny-ImageNet |
|---|---|---|---|---|
| Positive Sample Size $M$ | 4 | 4 | 4 | 4 |
| Positive Scaling $d_{\text{pos}}$ | 1 | 2.0 | 1.0 | 4.0 |
| Negative Scaling $d_{\text{neg}}$ | 1 | 0.5 | 10.0 | 1.0 |
| Negative Sample Size $N$ | 510 | 510 | 510 | 510 |
| Batch Size | 256 | 256 | 256 | 256 |
| Optimizer | Adam | Adam | Adam | Adam |
| Learning Rate | 1e-3 | 1e-3 | 1e-3 | 1e-3 |
| Weight Decay | 1e-6 | 1e-6 | 1e-6 | 1e-6 |
| Temperature Scaling $\tau$ | 0.5 | 0.5 | 0.5 | 0.5 |
| Feature Dimension | 128 | 128 | 128 | 128 |
| Data Augmentation | Fig 6 | Fig 6 | Fig 6 | Fig 6 |

```python
train_transform = transforms.Compose([
    transforms.RandomResizedCrop(32),
    transforms.RandomHorizontalFlip(p=0.5),
    transforms.RandomApply([transforms.ColorJitter(0.4, 0.4, 0.4, 0.1)], p=0.8),
    transforms.RandomGrayscale(p=0.2),
    GaussianBlur(kernel_size=int(0.1 * 32)),
    transforms.ToTensor(),
    transforms.Normalize([0.4914, 0.4822, 0.4465], [0.2023, 0.1994, 0.2010])])
```

Figure 6: PyTorch code for SimCLR data augmentation from Chuang et al. (2020).

In the SIMCLR framework, the number of negative samples is related to the batch size as follows:
$N = 2 \times (\text{Batch Size - 1})$

