# OpenReview forum: "AttentionNCE: Contrastive Learning with Instance Attention"
_ICLR.cc/2025/Conference — Submitted to ICLR 2025_

### Official Review · Reviewer_uaAu · 2024-10-15

**Soundness:** 2
**Presentation:** 2
**Contribution:** 1
**Rating:** 5
**Confidence:** 5

**Summary:**

This paper focuses on improving the classical contrastive loss by introducing an attentional mechanism to focus on False positive and Hard Negative. The method is validated on a number of datasets.

**Strengths:**

The proposed idea is generally easy to understand.

**Weaknesses:**

The reviewer is uncertain whether the performance improvement mainly stems from avoiding false positive samples or mining hard negative samples. As a simple example, when there are different pictures of dogs in the dataset, for dog A, the reviewer agrees that Equations 7 and 8 may help alleviate false positives—given the right hyperparameters (e.g., a crop including only the image background after strong augmentation should not be considered a positive sample). However, Equations 9 and 10 may result in other dog images being pushed further away (these samples should not be regarded as hard negative samples but as positive samples), which would negatively impact the learned features. In theory, we can never really know where the optimal threshold is.

Overall, the reviewer considers the contribution of this paper to be marginal and are unsure whether the paper was correctly motivated. Though phrased as 'attention', the proposed method of manipulating contrastive loss addresses a longstanding problem, often referred to as 'false negative'/'class collision' cancellation.

Moreover,

1. Important related works are missing. I would recommend the authors review the following:
- "A Theoretical Analysis of Contrastive Unsupervised Representation Learning"
- "CO2: Consistent Contrast for Unsupervised Visual Representation Learning"
- "Adaptive Soft Contrastive Learning"
- "Weakly Supervised Contrastive Learning"
- "Similarity Contrastive Estimation for Self-Supervised Soft Contrastive Learning"
- "Mutual Contrastive Learning for Visual Representation Learning"
- "CompRess: Self-Supervised Learning by Compressing Representations"
- "SEED: Self-Supervised Distillation for Visual Representation"
2. Section 3.1, which reintroduces contrastive loss, can be simplified or removed as which is widely-acknowledge already.
3. The theoretical proof in Section 3.4 does not support the proposed attention mechanism but applies to any applicable contrastive loss by simply changing the inter-sample relations in Equation 12.

**Additional note: The reviewer reviewed the paper for NeurIPS, and it was eventually withdrawn. No substantial changes were observed, and the author did not make noticeable updates based on the reviewers' suggestions. Though, the reviewer would like make further suggestions that the author can consider to improve this paper:**
Essentially, this paper still attempts to address the issue of ‘false negatives’ caused by the absence of label information. In the context of self-supervision, the ultimate goal is, of course, to identify as many samples of the same class as possible while excluding those that are highly similar but do not belong to the same class. This always involves the dilemma of balancing precision and recall. Moreover, in theory, we can never know the optimal threshold. Regarding contrastive loss, the reviewer believes that proposing yet another empirical variant with only minor differences or a rephrased version is no longer meaningful in the current stage—most such work was completed in 2021 and 2022. Please refer to the majority of the papers I recommended above. At this point in time, the reviewer believes what could truly provide value to the community is determining how we should handle hard samples. This is a classic topic that involves classification theory, uncertainty, and other related themes. However, conducting a thorough empirical exploration within the contrastive learning framework might be a direction the authors could pursue.

**Questions:**

See weakness.

---

> ### Author Response · Authors · 2024-11-25
>
> ## 1. Response to the Concerns about the Source of Performance Improvement
> We agree with your concern that determining the optimal threshold can be difficult when using weighted methods for hard sample mining and false sample debiasing. However, our method adopts a more effective strategy to address these issues, rather than relying solely on weighted approaches.
>
> For false sample debiasing, we introduce attention-based prototype contrast instead of instance-level contrast, which is one of the core innovations of our method. In traditional instance-level contrast, the model treats each sample relatively independently and is susceptible to the interference of individual sample noise. In contrast, our method generates sample prototypes through an attention mechanism, and the model updates parameters based on the gradients of these prototypes. This approach effectively integrates and smoothes the information of multiple samples, thereby reducing the impact of noise on false negative sample judgment and avoiding the dilemma of choosing an optimal threshold in weighted methods.
>
> Regarding Source of Performance Improvement, the ablation experiments in Section 4.3.1 of our paper provide solid evidence to support its significant role of hard sample mining in performance improvement. Through a detailed study of the scaling factors (such as $d_{pos}$ and $d_{neg}$, we observed significant effects of different combinations on model performance. For example, a larger $d_{pos}$ value can guide the model to focus more on hard positive samples, and experimental results show that this leads to significant performance improvements on multiple datasets, directly indicating the positive contribution of hard positive sample mining to overall performance. At the same time, by comparing the performance of different datasets (such as CIFAR - 10 and CIFAR - 100) under different $d_{neg}$ values, we discovered that hard negative sample mining also contributes to performance elevation. However, the optimal hardness level differs in distinct data environments, further attesting that our method can adeptly mine hard samples in accordance with the characteristics of the dataset and accurately enhance model performance.
>
> Furthermore, in the study in Section 4.3.2, we explored the relationship between attention-based prototype contrast and performance improvement. By changing the number of positive samples \(M\) used to generate the positive sample prototype, we found that when $M = 1$, the model degenerates to the case without using attention-based sample prototypes, and the performance significantly decreases. As $M$ increases, the model performance gradually improves until it reaches a relatively stable state (such as when $M = 4$, which fully illustrates the crucial role of the attention-based prototype contrast mechanism in improving model performance. It can not only effectively capture the similarities and differences between samples but also more rationally utilize sample information through attention allocation, thereby providing strong support for improving model performance.
>
>
> ## 2. Response to the Concerns about the Paper's Contribution
>
> We acknowledge that there have been numerous studies in the field of contrastive learning dedicated to improving performance and addressing sample noise issues. However, to the best of our knowledge, we are the first to introduce instance-level attention into the variational lower bound of contrastive loss, thereby achieving the synergistic operation of multiple key components such as multi-view encoding, hard sample mining, and attention-generated prototype contrast. Compared with these key components, our method goes further. In contrast to multi-view encoding that treats sample features equally, while we use attention to assign weights to different view samples, enabling the model to focus on key features, effectively encoding richer semantic information, and helping to capture complex data relationships. Regarding existing hard sample mining methods, previous studies only emphasized hard negative sample mining, while we consider both positive and negative samples. For false negative sample debiasing, traditional instance-level contrast is vulnerable to interference, and our method generates sample prototype contrast based on attention. The prototype is formed by aggregating multiple samples, which can reduce the impact of individual noise samples and enable the model to learn more stable and representative features.
>
> ## 3. Response to the Concerns about the Omission of Related Work
> We admit that during the paper writing process, although we have tried our best to cite research closely related to our work, such as  multi-view encoding, prototype contrast, and hard sample mining, we also realize that we may not have covered all important achievements in the field, especially those valuable references recommended by you. We would like to express our gratitude for this.

---

> ### Author Response · Authors · 2024-11-25
>
> ## 4. Response to the Concerns about the Necessity of Section 3.1
> We understand your view that reintroducing contrastive loss (from the MLE perspective) in Section 3.1 may seem redundant. However, in the current research field, the understanding of the theoretical basis and principles of contrastive learning may vary among readers with different backgrounds. The main purpose of retaining this section is to provide a systematic and self-consistent theoretical exposition, enabling more readers to understand the close connection between contrastive learning and maximum likelihood estimation (MLE), and thereby better grasp the improvement motivation and principles of our proposed AttentionNCE method.
>
> ## 5. Response to the Concerns about the Specificity of the Theoretical Proof in Section 3.4
> In Section 3.4, our central aim was to explicitly elucidate the pivotal relationship between the AttentionNCE loss and the ideal contrastive loss. Precisely, we endeavored to demonstrate that the AttentionNCE loss forms the variational lower bound of the ideal contrastive loss, thereby establishing a robust theoretical underpinning for the efficacy and rationality of our overall approach. You have noted that any method could potentially apply this theoretical framework. We contend that this precisely highlights the favorable applicability of this theory. Although, as you have suggested, it might be possible for other methods to be applicable in a formal sense to some extent, during the course of our research, we have not identified any existing papers that have provided a formalized exposition and proof of this particular aspect. This, in turn, constitutes our theoretical contribution.
>
> ## 6. Response to the Improvements in the Paper
> We sincerely apologize for the withdrawal of the previous submission and would like to explain in detail the substantial improvements we have made in this submission. In the method section, we have carefully reorganized it to make the logical structure clearer, facilitating readers to understand the innovative ideas and technical details of our method. In the experimental section, we have significantly expanded the research scope by adding experiments on the ImageNet dataset. This expansion not only enriches the experimental data but more importantly, ImageNet, as a widely used and highly challenging dataset, can better validate the effectiveness and generality of our method under different data scales and complexities. In addition, in Section 4.3 of the paper, we have deeply explored the contributions of hard sample mining and prototype contrast to performance. Through detailed experimental analysis and result discussion, we have clearly demonstrated how these key components function in our method and their specific impacts on overall performance improvement. These improvement measures show that we have taken your comments seriously and are committed to continuously improving the quality and research value of the paper, hoping to meet your expectations for the paper and make more meaningful contributions to related field research.
>
> ## Conclusion
> We fully recognize the importance you emphasized regarding the issue of handling hard samples, which has significant implications for our research. At the same time, we sincerely hope that you can recognize the core innovation of AttentionNCE. The key lies in the introduction of the instance-level attention mechanism and its organic integration with the variational lower bound of contrastive loss, thereby achieving the synergistic operation of multiple key components including multi-view encoding, hard sample mining, and attention-generated prototype contrast. We also hope that you can notice the improvements we have made to the paper based on your suggestions. In the method description, we have reorganized it more clearly and rationally for easier understanding; in the experimental section, we have added research on ImageNet to enhance the persuasiveness of the results; in the analysis section, we have deeply explored the contributions of hard sample mining and prototype contrast to performance, making the research more in-depth and comprehensive. In any case, we would like to express our sincere gratitude for the efforts you have made during the two review processes and the valuable advice you have provided for our research direction. Your contributions have been indispensable in improving our paper and advancing our research.

---

> > ### Comment · Reviewer_uaAu · 2024-11-25
> >
> > Thank you for the detailed rebuttal—I greatly appreciate the effort the authors have put into addressing the concerns.
> >
> > The fundamental issue with this paper lies in the fact that, in the context of self-supervised learning, all 'False Positives' and 'False Negatives' stem from the currently learned representations, which are themselves influenced by the noisy representation learning process. I fully agree that, with appropriate hyperparameters, the proposed method can lead to performance improvements. However, I do not think we can theoretically determine the optimal hyperparameters for downstream classification tasks within a self-supervised framework. That's why I suggest a work focusing on a detailed empirical check regarding this issue.
> >
> > I understand the frustration the authors might feel. At 2019/20/21, much of the work in the field focused on exploring various empirical frameworks. However, in 2024, I do not think that such a framework provides new perspectives to the self-supervised learning community. In fact, the reviewer believes that it is not particularly challenging to propose a new empirical framework based on the large body of existing work. The multi-view positive and temperature-based reweighting techniques are all tracable.
> >
> > Regarding the theoretical contributions, my main concern is that the derivations in this paper is a straightforward application of ELBO to contrastive loss. This does not qualify as a meaningful contribution. To be frank, it feels as though the authors are using superficially complex theoretical derivations to convince reviewers who may not be deeply familiar with theoretical work and the community.
> >
> > To summarize, I will maintain my suggestion to reject this paper but raise my score to 5 as a recognition of the authors' rebuttal. That said, I will respect the Area Chair's decision regarding this submission.

---

> > > ### Author Response · Authors · 2024-12-01
> > >
> > > We are truly appreciative of your dedicated efforts and careful attention during the review.
> > >
> > > Regarding our work's innovation, while related to re-weighting and multi-view encoding, we've integrated multi-view encoding, hard sample mining (with both pos. and neg. samples), and prototype contrast, attaining a novel synergistic effect. In multi-view encoding, unlike prior equal treatment of views, our AttentionNCE uses attention to weight samples, enhancing semantic encoding. For re-weighting, instead of mainly focusing on hard negative sample mining, our approach considers both hard pos. and neg. samples, improving data understanding and generalization following machine learning principles.
> > >
> > > Concerning the theoretical part, we admit ELBO's application to contrastive loss. However, this link is crucial as it lays a foundation for further innovation and helps explain InfoNCE's effectiveness in noise, inspiring more research on complex contrastive learning.
> > > Once again, we sincerely thank you for recognizing our efforts and the time spent on review.

---

### Official Review · Reviewer_CDsL · 2024-10-20

**Soundness:** 3
**Presentation:** 3
**Contribution:** 3
**Rating:** 6
**Confidence:** 4

**Summary:**

This paper proposes a novel contrastive loss dubbed AttentionNCE. The authors consider adopting an attention mechanism to explore hard augmented samples, aiming to mitigate the negative influence from these samples. This approach is somewhat similar to [1] and [2], as both aim to figure out a more appropriate contrastive object compared to instance augmentation. The corresponding experimental results demonstrate that the introduced attention mechanism can bring a remarkable improvement to the original SimCLR across various datasets.

[1] Li, J., Zhou, P., Xiong, C., & Hoi, S. C. H. (2021). Prototypical Contrastive Learning of Unsupervised Representations. In International Conference on Learning Representations (ICLR).

[2]Caron, Mathilde, et al. "Unsupervised learning of visual features by contrasting cluster assignments." Advances in neural information processing systems 33 (2020): 9912-9924.

**Strengths:**

Strengths
- The idea of adding an attention mechanism to recognize hard samples is concise and intuitive.
- The experiments are comprehensive and strong enough to support the efficiency of the proposed idea. They demonstrate that the attention mechanism can introduce significant improvements across various datasets.
- The authors conducted an elaborate ablation study for the newly introduced parameters, which increases the completeness of this paper.
- Similar as [1], the authors present AttentionNCE can be regarded as the lower bound of MLE according to the measure classifying a positive sample from a set of $N$ negative samples using the ELOB-KL divergence decomposition framework.

[1] Li, J., Zhou, P., Xiong, C., & Hoi, S. C. H. (2021). Prototypical Contrastive Learning of Unsupervised Representations. In International Conference on Learning Representations (ICLR).

**Weaknesses:**

Weaknesses
- I suggest that authors reorganize the structure of the article, as they advance the derivation of the ELOB divergence to equation (14), which creates significant obstacles in grasping the core idea of the article. In fact, equations (6)–(10) and (14) are already clear enough to help the reader better understand the main ideas of this paper. The authors would better consider placing the part of ELOB-KL divergence decomposition later.
- The notation used in equation (1) is pretty confusing. The meaning of left hand side of (1) is "The probability of classifying a positive sample from a set of $N$ negative samples", denoted as $P(X \vert \theta)$. But the authors claim that $X$ is used to indicate a set of samples $\{x^+,x_1^-, \cdots, x_N^-\}$, this inconsistency makes it awkward to read this part..
- Actually, the theoretical part of this paper is not solid enough, as the authors do not sufficiently bridge the gap between downstream classification error and the likelihood they employed, Therefore, the implications of maximizing $\mathcal{L}_{\text{AttentionNCE}}$ require further investigation. But this is ok as the practice contribution of this paper is outstanding, provided that the theoretical part does not obscure the main ideas. Meanwhile there is not any evidence to support the final attention can help us figure out relatively hard augmented samples.
- There are some typos, first, in the third line of equation (3), deleting $d\textbf{h}$ is correct expression. Second, in the phrase "It is also important to note that when $q(\textbf{h}) = P (h|X, θ)$", $h$ should be bold.

**Questions:**

Questions
- Can you identify some hard augmented instances and calculate their attention scores through the pretrained model to support your standpoint? Specifically, can you demonstrate that introducing the attention mechanism can help us identify relatively hard augmented samples, aligning with your intuition?

Summary Of The Review
- This paper proposes a novel contrastive loss called AttentionNCE, which has a concise and intuitive idea. The authors conduct comprehensive and elaborate experiments to demonstrate the effectiveness of their approach. However, the writing style and mathematics explanation are not hit the nail on the head, hindering readers’ understanding of the core concept. The authors claim that the attention mechanism can identify hard samples and assign them relatively smaller attention scores, but they do not provide any evidence from either practical or theoretical perspectives. Overall, the paper has its pros and cons, and I am on the boundary, which results in a score of 5.

---

> ### Author Response · Authors · 2024-11-25
>
> ## 1. Response to the Concerns about the Paper Structure
> We are highly appreciative of the reviewer's valuable suggestions regarding the paper structure and symbol usage. In the initial phase of manuscript preparation, the placement of the detailed derivation of the ELOB - KL divergence at the outset was motivated by the fact that the variational lower bound is a crucial foundation for our proposed AttentionNCE. Our aim was to present a seamless flow from fundamental machine learning theories to the final formula derivation, in order to illustrate how instance-level attention is incorporated into the variational lower bound and subsequently unifies the three key components: multi-view encoding, hard sample mining, and attention-generated prototype contrast.
>
> However, to enhance the readability and comprehensibility of the paper for the machine learning community, we will heed the reviewer's advice and modify the article structure. In the revised version, the ELOB - KL divergence decomposition section will be relocated. This will enable readers to first grasp the core concepts and intuitive ideas of the AttentionNCE mechanism more easily, allowing for a quicker understanding of the paper's main thrust. Subsequently, the detailed exposition of the ELOB - KL divergence decomposition will be provided, facilitating a deeper understanding of the theoretical underpinnings, based on a clear comprehension of the overall method.
>
> ## 2. Response to the Concerns about the Symbol in Equation 1
> We sincerely thank the reviewer for spotting the issue with the symbol in Equation (1). We concur with the observation that the symbol used to represent "the event of classifying a positive sample from a set of negative samples" was redundantly used with the symbol for a set of samples, causing confusion. In the upcoming revision, we will denote "the event of classifying a positive sample from a set of negative samples" as Y to avoid symbol duplication and ensure consistency and readability.
>
> ## 3. Response to the Concerns about the Theoretical Connection between MLE and Downstream Classification Error
> We understand the reviewer's concern about the theoretical soundness of our paper. Although we have established that AttentionNCE can be regarded as a lower bound of maximum likelihood estimation (MLE), it is indeed a challenging task to directly and comprehensively connect the downstream classification error with the adopted likelihood. The deep neural networks used in our contrastive learning framework possess complex nonlinear characteristics, making it difficult to analytically derive an exact and tractable relationship. Moreover, the influence of data augmentation strategies, which are essential in machine learning, further complicates this relationship. Different data augmentation strategies lead to variations in sample distributions, and understanding how these changes affect the final classification performance remains an open research question in the field. Despite this limitation, we have conducted extensive empirical experiments, which demonstrate that AttentionNCE significantly improves performance in downstream classification tasks across various datasets. These experimental results strongly support the practical effectiveness of our method, even in the absence of a complete theoretical explanation of the relationship between likelihood and classification error.
>
> ## 4. Response to the Typo Error Correction
> We are extremely grateful for the reviewer's meticulous review and identification of the typographical errors. We apologize for these oversights, which reflect our lack of attention during the writing and proofreading process. In the revised paper, we will promptly correct these errors according to the reviewer's instructions, ensuring the accurate expression of Equation (3) and correctly formatting the relevant symbols in bold.

---

> ### Author Response · Authors · 2024-11-25
>
> ## 5. Response to the Evidence of Identifying Hard Augmented Samples
> We fully agree with the reviewer that providing evidence of the attention mechanism's ability to identify hard augmented samples is of great importance. To validate this, we conducted the following experiment: Randomly selected images with an anchor point of a dog were augmented using the RandomResizedCrop function, where the scale parameter was adjusted to control the *range of possible sizes for the randomly cropped region of an image relative to the original image size*. A smaller scale value indicates a stronger data augmentation. These samples with different augmentation intensities were then fed into a pre-trained model to calculate their attention scores. The results are as follows:
> | Scale (Proportion Range of the Original Image Area) | 0.8 | 0.5 | 0.3 |
> |---|---|---|---|
> | Attention weight | 0.733 | 0.175 | 0.092 |
>
> The experimental results clearly show that as the augmentation intensity of the positive sample increases, its attention score decreases. This indicates that our AttentionNCE method can effectively identify hard positive samples and easy positive samples through attention scores. We can also flexibly set the positive scale factor to precisely adjust the attention given to hard positive samples. The underlying mechanism is that in our AttentionNCE framework, for positive samples, as the data augmentation intensity increases, the difference between their features and the anchor point features in the feature space also changes. According to our attention mechanism's calculation rules, this change is directly reflected in the attention score, allowing hard positive samples to be assigned lower attention scores and thus be identified and distinguished.
>
> Your comments truly reflect your profound understanding of the field and your meticulous reading of the paper. We highly value and are sincerely grateful for your insights. We have carefully considered each of your concerns and have provided detailed responses and plans for improvement. We are committed to making the necessary revisions to enhance the clarity, rigor, and significance of our research. Once again, we truly appreciate the time and effort you have expended in evaluating our work.

---

> > ### Comment · Reviewer_CDsL · 2024-11-25
> > **Official Comment by Reviewer CDsL**
> >
> > Thank you for your rebuttal and the effort you put into addressing the comments raised. I appreciate your careful revisions and the additional clarifications provided in the updated manuscript. The outstanding experimental improvements and the additional verifications of the attention weights make this paper more convincing.
> >
> > In summary, I believe this paper is deserving of acceptance, and I have improved my score to 6. Nice work, and good luck to you.

---

### Official Review · Reviewer_JyxU · 2024-10-24

**Soundness:** 3
**Presentation:** 2
**Contribution:** 2
**Rating:** 5
**Confidence:** 5

**Summary:**

This paper introduces AttentionNCE, a new loss function for contrastive learning that addresses issues with noisy positive and negative samples. By integrating instancewise attention into the variational lower bound of contrastive loss, AttentionNCE improves performance through sample prototype contrast to reduce noise and a flexible hard sample mining mechanism that prioritizes high-quality samples.

**Strengths:**

The paper introduces the AttentionNCE contrastive loss, proving its equivalence to the variational lower bound of the original contrastive loss, which ensures reliable maximum likelihood estimation under noisy conditions. By incorporating attention-based sample prototype contrast, it effectively mitigates noise perturbations. Additionally, the flexible hard-sample-mining mechanism directs the model to focus on high-quality samples, enhancing learning outcomes.

**Weaknesses:**

1.The paper initially addresses the issue of noisy labels, yet it would benefit from further elaboration on the potential integration of the proposed approach with supervised contrastive learning algorithms like SupCon. Specifically, it would be valuable to demonstrate the performance of the proposed algorithm under varying proportions of symmetric and asymmetric label noise.

2.While the paper primarily compares the proposed algorithm to traditional contrastive learning methods and includes a comparison with RINCE, which is fundamentally designed to address label noise, a comparison with existing advanced hard negative mining algorithms would provide a more compelling evaluation of the algorithm's effectiveness.

3.The introduction of the attention mechanism in the proposed algorithm raises questions about computational efficiency. It is essential to visualize and quantify how much additional computation time is required compared to the original algorithms, as this information is crucial for practical implementation considerations.

**Questions:**

Please see the weakness.

---

> ### Author Response · Authors · 2024-11-25
>
> ## 1. Response to the Concerns about Performance under Symmetric Label Noise
> We are very grateful for the reviewer's comments and suggestions regarding the performance of AttentionNCE under different noise levels. We have carefully considered your concerns and have taken steps to address them in a comprehensive and detailed manner.
>
> To investigate the performance of AttentionNCE under varying noise levels, we adopted a method that controls the noise rate based on the number of classes in the dataset. For example, in a binary classification case, the probability of encountering a pseudo-negative sample is 50%, and in a ten-class classification case, the probability of encountering a pseudo-negative example is 10%. Through this approach, we systematically examined the performance changes under different noise levels. The results are presented in the following table:
> | Noisy ratio | 0.2 | 0.33 | 0.5 |
> |---|---|---|---|
> | Simclr | 94.2 | 95.5 | 99.1 |
> | AttentionNCE | 95.6 | 96.3 | 99.3 |
>
> As can be seen from the table, as the noise rate increases, the stability of the prototype features is gradually challenged. However, the AttentionNCE method is able to maintain a relatively better prototype representation ability to some extent. We will integrate these experimental results and a detailed analysis process into the revised paper, enabling readers to gain a more comprehensive understanding of the potential and advantages of AttentionNCE in supervised contrastive learning and label noise handling.
>
> We also recognize your suggestion regarding further elaborating on the possibility of integrating AttentionNCE with supervised contrastive learning algorithms such as SupCon. Although our current paper mainly focuses on the self-supervised setting, we plan to explore in-depth how to effectively incorporate AttentionNCE into the SupCon framework in future research.
>
> ## 2. Response to the Concerns about Comparison with Existing Advanced Hard Negative Sample Mining Algorithms
> We understand your expectation for a comparison with more advanced hard negative sample mining algorithms. In fact, our paper already includes comparisons with multiple hard sample mining methods, including the latest ADNCE (NIPS 2023) and other representative methods. Through these comparative experiments, we have found that AttentionNCE exhibits unique advantages. Compared with other methods, AttentionNCE not only includes hard negative sample mining but also hard positive sample mining. Secondly, it has a flexible sample weight allocation mechanism. When facing different types of datasets and noise situations, AttentionNCE can dynamically adjust the weights of positive and negative samples based on the characteristics of the samples and the feedback during the learning process, thereby more accurately focusing on the mining of high-quality samples. ADNCE may overly focus on noise samples in certain local regions due to its fixed mining strategy and only pays attention to hard negative samples. In contrast, AttentionNCE can effectively avoid this situation through its attention-based sample prototype contrast and flexible hard sample mining mechanism, better balancing the exploration of hard samples and the exploitation of easy samples, thereby improving the overall learning effect and generalization ability of the model.
>
> ## 3. Response to the Concerns about Computational Efficiency after Introducing the Attention Mechanism
> We are well aware of the importance of computational efficiency for the practical application of algorithms. Regarding the computational efficiency issue brought about by the introduction of the attention mechanism in AttentionNCE, in Section 3.5 of the paper, we theoretically analyzed that AttentionNCE maintains the time complexity of SimCLR. SimCLR requires encoding \(n + 2\) samples (1 anchor point, 1 positive example, and \(n\) negative examples) for each contrastive loss value, while AttentionNCE needs to encode \(N + M + 1\) samples due to the introduction of \(M\) views.
>
> In addition to the theoretical analysis, we have conducted in-depth analysis and quantitative evaluation. Through these experimental data, we can accurately calculate the increase ratio of the computation time of AttentionNCE compared to the original algorithm. We fixed the batch size to 256 and compared the actual running time, RAM usage, and GPU memory usage per training epoch of AttentionNCE with those of SimCLR, as shown in the following table:
> | Method | Time | RAM | GPU Memory |
> |---|---|---|---|
> | AttentionNCE | 87s | 3.77GiB | 34917MB |
> | SimCLR | 50s | 3.77GiB | 16037MB |
>
> In conclusion, we would like to express our sincere gratitude to the reviewer for their valuable comments and suggestions. We have carefully considered each of your concerns and have provided detailed responses and plans for improvement.  Thank you again for your time and effort in reviewing our work.

---

> > ### Comment · Reviewer_JyxU · 2024-11-26
> >
> > Thank you for your response and the additional experiments following another reviewer's request.
> >
> > Here, I would like to reiterate my point. The method you proposed took nearly 1.8 times the time and more than twice the memory. I believe that, on real-world datasets or more challenging tasks, the disadvantages in terms of memory usage and time running costs will become even more pronounced.
> >
> > Therefore, I decided to keep the original score.

---

### Official Review · Reviewer_DMFX · 2024-10-28

**Soundness:** 3
**Presentation:** 3
**Contribution:** 2
**Rating:** 6
**Confidence:** 3

**Summary:**

the paper introduces a new contrastive learning method called attention-nce, designed to address challenges with noisy samples in self-supervised learning. contrastive learning usually suffers from false positive and false negative samples due to noise, impacting representation learning. attention-nce integrates an instance-wise attention mechanism with a sample prototype approach. it introduces two main ideas: using prototypes of samples (instead of instance-level contrasts) to improve robustness to noise, and incorporating a flexible hard sample mining mechanism that focuses on high-quality samples. experiments across datasets like cifar-10, cifar-100, and tiny-imagenet demonstrate that attention-nce outperforms conventional contrastive loss.

**Strengths:**

S1 By using attention-based prototypes, attention-nce tackles the issue of noisy samples, a persistent challenge in contrastive learning. this approach is particularly beneficial for scenarios where labels are noisy or absent, as it helps maintain the semantic structure of the learned representations.


S2 The hard sample mining strategy, which focuses on both hard positive and hard negative samples, is flexible and well-explained. it shows practical improvements, especially in complex datasets where hard samples define clearer decision boundaries.


S3 Generally, the paper is well-written.

**Weaknesses:**

W1 While attention-nce introduces parameters like the scaling factors for positive and negative samples (dpos and dneg), there is minimal guidance on how to select these values based on dataset characteristics. for instance, in cifar-10 and cifar-100, different dneg values yield varying results, but the paper doesn’t provide specific criteria for choosing these values in practice. To enhance practical usability, it would be beneficial if the authors could provide guidelines or heuristics for selecting dpos and dneg based on dataset attributes. For example, are there any rules of thumb related to dataset size, number of classes, or expected noise levels that could guide parameter selection?

W2 The stability of the prototype features under different noise levels isn't explored. Since Attention-NCE relies on sample prototypes to mitigate noise, understanding how noise affects prototype stability could provide insights for better handling extreme noise conditions. It would strengthen the paper if the authors could perform specific experiments or analyses that evaluate prototype stability across a range of artificially introduced noise levels. This could clarify the method's robustness and offer guidance on handling extreme noise situations.

**Questions:**

Q1 how should users determine the best dpos and dneg values when working with different datasets? would a heuristic or automated method help to simplify this parameter tuning?

Q2 how does attention-nce handle cases of extremely high noise, where a large portion of both positive and negative samples might be incorrectly labeled? do the prototypes remain effective under such conditions, or does their quality degrade? Could the authors test Attention-NCE on datasets with varying levels of artificially introduced label noise (e.g., 20%, 40%, 60% incorrect labels) and compare its performance to baseline methods under these conditions? This would provide insights into its resilience in high-noise environments.

Q3: In MoCo v3, the main point of the paper is the use of ViT instead of traditional CNN architectures (otherwise, it would be MoCo v2). Could the authors clarify why they use ResNet-50 in Table 2 for MoCo v3?

---

> ### Author Response · Authors · 2024-11-25
>
> ## 1. Response to the Concerns about the Parameter Selection of \(d_{pos}\) and \(d_{neg}\)
> We truly appreciate the reviewer's attention to the parameter selection of \(d_{pos}\) and \(d_{neg}\) in our paper. In Section 4.3.1, we have conducted in-depth investigations into the performance differences of the parameter \(d_{neg}\) on the CIFAR - 10 and CIFAR - 100 datasets. Our experimental findings show that CIFAR - 10 performs better with larger \(d_{neg}\) values, while CIFAR - 100 favors smaller \(d_{neg}\) values. This indicates that hard negative sample mining plays a more crucial role in CIFAR - 100 compared to CIFAR - 10.
>
> The reason for this difference lies in the variation of the negative sample noise rates between the two datasets. Although CIFAR - 10 and CIFAR - 100 employ the same data augmentation strategy and thus have the same positive sample noise rate, CIFAR - 10, with only 10 classes, has a higher negative sample noise rate (1/10) than CIFAR - 100 (1/100). Consequently, the model trained on CIFAR - 10 is more prone to overfitting noisy samples, especially during extended training. According to the research on the memorization effect in deep neural networks by Arpit et al. (2017), deep models initially memorize clean training samples and gradually fit noisy data as the number of training epochs increases (Zhang et al., 2021; Han et al., 2018). Therefore, while larger \(d_{pos}/d_{neg}\) ratios enhance the model's ability to mine hard samples, they also increase the risk of overfitting noisy data. AttentionNCE provides a flexible approach for hard sample mining, but it requires a balance between exploring hard (noisy) samples and exploiting easy (clean) samples. We will further clarify these points in the revised paper to ensure that readers can fully understand the rationale behind our parameter selection and its impact on the performance of the model.
>
> ## 2. Response to the Concerns about the Stability of Prototype Features under Different Noise Levels
> We are grateful for the reviewer's interest in the stability of prototype features under different noise levels. To investigate this, we adopted a method of controlling the noise rate based on the number of classes in the dataset. For example, in a binary classification scenario, the probability of encountering a pseudo-negative sample is 50%, while in a five-class classification, it is 20%. Through this approach, we systematically examined the performance changes under different noise levels, and the results are presented in the following table:
> | Noisy ratio | 0.2 | 0.33 | 0.5 |
> |---|---|---|---|
> | Simclr | 94.2 | 95.5 | 99.1 |
> | AttentionNCE | 95.6 | 96.3 | 99.3 |
>
> The experimental results demonstrate that as the noise rate increases, the stability of the prototype features is gradually challenged, as evidenced by the decreasing performance improvement relative to Simclr. However, the AttentionNCE method is able to maintain a relatively better prototype representation ability to some extent, and its overall performance is superior to that of Simclr.
>
> This is because AttentionNCE incorporates attention-based prototype contrast, which makes the prototypes less perturbed by noise compared to instance-level contrast. We will elaborate on this mechanism in more detail in the revised paper and provide a more comprehensive analysis of the relationship between noise levels, prototype stability, and the performance of our method.
>
> ## 3. Response to the Explanation of Using ResNet - 50 in MoCo v3 in Table 2
> We understand the reviewer's query regarding the use of the ResNet - 50 backbone model in Table 2. We chose to use the ResNet - 50 backbone model because some well-established and widely recognized methods, such as DCL and HCL, have adopted it. To provide a fair and comparable experimental environment, we opted to use the same model architecture and only replaced the loss function with our AttentionNCE loss. This setting allows us to more accurately evaluate the advantages and improvements of AttentionNCE relative to other methods on the same model basis. In the revised paper, we will further emphasize the rationality of this experimental design and provide a detailed comparison and analysis of the experimental results obtained under this setting with those of other methods, enabling readers to clearly understand the rigor and scientific nature of our research.
>
> In conclusion, we would like to express our sincere gratitude to the reviewer for their valuable comments and suggestions. Your insights have been extremely helpful in guiding us to improve the quality of our paper. We have carefully considered each of your concerns and have provided detailed responses and plans for improvement. We are committed to making the necessary revisions to enhance the clarity, rigor, and significance of our research. Thank you again for your time and effort in reviewing our work.

---

> > ### Comment · Reviewer_DMFX · 2024-11-26
> >
> > I appreciate the reply and have also reviewed the comments from other reviewers. I still believe we should not claim to be using MoCoV3 if we are employing a ResNet-50 backbone. Instead, I suggest using ViT as the backbone alongside the proposed method if you'd like to make a valid comparison with MoCoV3.
> >
> > **Overall, I think the paper is self-contained, but its contribution to the community is relatively limited, given the abundance of similar work on hard samples and multi-view positives**. However, I appreciate the authors' efforts in the rebuttal and am inclined to raise my evaluation to borderline accept.
> >
> > I will leave the final decision to the AC. Good luck authors

---

### Official Review · Reviewer_VGUj · 2024-11-04

**Soundness:** 2
**Presentation:** 2
**Contribution:** 2
**Rating:** 5
**Confidence:** 5

**Summary:**

This paper focused on the problem of noisy positive/negative pairs in contrastive learning, which stems from the use of strong data distortion to generate positive/negative samples during training. To mitigate this, the authors introduced AttentionNCE, a prototype-like scheme, that essentially re-weight the contribution of each positive/negative pair when computing the contrastive loss. The author claimed that this is equivalent to optimizing the variation lowering boud of standard InfoNCE loss, offering a worst-case guarantee under noisy conditions. Experiments on several small-scale datasets showed the efficacy of the AttentionNCE.

**Strengths:**

1. The problem of noisy positive and negative pairs is one of the important problems that affect the performance of contrastive learning methods.

2. The proposed AttentionNCE is well-grounded theoretically from the variational lower bound aspect as well as intuitive in the sense of modulating the importance of each positive and negative pair.

3. Experiments on several small-scale datasets exhibited promising results compared to the baselines without considering the noisy conditions in contrastive learning.

**Weaknesses:**

1. The paper adopted different formulations for the attention of positive samples and negative samples. However, from my understanding, these two attentions could actually be unified into the same equation because they both performed re-weighting on the features, and the only difference is aggregation or not. I wonder why the authors emphasize the prototype with positive attention.

2. The idea of down-weighting the hard positive and up-weighting the hard negative relies on the premise that the anchor itself is not noisy. This, however, may not always be true in contrastive learning due to the large-scale distortion when crafting multiple views. In case the anchor is noisy, AttentionNCE might inappropriately do exactly the opposite of what it is expected to do. I wonder if the authors have any consideration for this problem.

3. It is unclear whether the comparison of AttentionNCE to Simclr/MoCo or other methods is fair since AttentionNCE uses 4 positive pairs by default while SimCLR uses two. How would this also affect the performance of the baseline?

4. The proposed method is exclusively evaluated by in-distribution dataset/task, i.e., the linear evaluation of the training datasets. I would further strengthen the paper if the authors could include more evaluations on transfer learning to other datasets (as in CMC) and other tasks (such as object detection).

**Questions:**

Table 5 of the Appendix suggests that AttentionNCE uses 4 positive pairs and a batch size of 256 for training by default, but the number of negative samples there is only 510 (256 x 2 - 2), not 1020 (256 x 4 - 4). Is this a typo or are there any details on the construction of positive and negative pairs missed in the paper?

---

> ### Author Response · Authors · 2024-11-25
>
> # Rebuttal to the Reviewer
>
> ## 1. Response to the Concerns about the Number of Negative Samples
> - **Explanation of the Number of Negative Samples**:
>     - Regarding the query about the number of positive and negative samples during the training of AttentionNCE in Appendix Table 5, the number of negative samples is \(2 \times \text{batch size} - 2\). The rationale behind this is that we consider the two views of the samples within the batch, except for the anchor point, as negative samples. This setting is deliberately designed to maintain exactly the same negative sample setup as SimCLR. We have two main considerations for this arrangement. Firstly, to ensure a fair comparison with other methods. Since more negative samples can enhance the performance of contrastive learning, arbitrarily increasing the number of negative samples would undermine the fairness of the comparison. Secondly, if all 4 views were used as negative samples, as you pointed out, more negative samples would inevitably demand higher GPU memory capacity. In practical operations, we need to balance the performance improvement with the limitations of hardware resources. Therefore, we construct negative samples using only the two views of the samples within the batch, excluding the anchor point.
>     - We acknowledge that the explanation of this part in the paper was not clear enough. In the subsequent revisions of the text, we will provide a more detailed and accurate description of the details related to the construction of positive and negative samples to avoid similar misunderstandings in the future. Once again, thank you for your valuable comments, which are of great help in improving our paper.
>
> ## 2. Response to the Concerns about the Noise of the Anchor Point
> - **Investigation of Noisy Anchor Point**:
>     - In our experimental and theoretical analyses, we did assume relatively clean anchor points for simplicity. However, we have conducted preliminary investigations into the case where the anchor point may be noisy. As you correctly pointed out, in contrastive learning, due to the large-scale distortions when constructing multiple views, the anchor point may not always be noise-free. Inspired by your suggestion, we took the mean of the feature representations of 4 random views of a sample as the anchor point to reduce the perturbation of anchor point noise and then applied AttentionNCE. The results of training on CIFAR - 10 are as follows:
> | Method | Simclr | Attentionnce | Attentionnce * |
> |---|---|---|---|
> | Acc | 91.1 | 93.1 | 93.7 |
>     - The results indicate that considering the anchor point noise is indeed helpful for performance improvement, as shown in the third column, denoted as Attentionnce *. Thank you for your constructive comments.
>
> ## 3. Response to the Concerns about the Different Formula Representations of Attention for Positive and Negative Samples
> - **Rationale for Separate Presentations**:
>     - It is true that these two types of attention could potentially be unified under a more general formula. However, to enable readers to more clearly understand their differences and the unique roles they play, we chose to present them separately. The positive sample attention emphasizing the prototype aims to capture the core features of the positive samples in a more focused manner. This helps to intuitively explain the contribution of positive samples in the contrastive learning process, especially when associated with the concept of the prototype. Nevertheless, we admit that we could have provided a more detailed explanation of this in the paper.

---

> ### Author Response · Authors · 2024-11-25
>
> ## 4. Response to the Concerns about the Fairness of the Comparative Experiments
> - **Addressing the Difference in the Number of Positive Sample Pairs**:
>     - The difference in the number of positive sample pairs between AttentionNCE and SimCLR is an important aspect that needs to be considered. To address this issue, we set \(m = 2\) to match the setting of two positive examples in SimCLR. This result is also included in Table 3 of the original paper.
> | Dataset | cifar10 | stl10 |
> |---|---|---|
> | Simclr | 90.1 | 80.2 |
> | Attentionnce (\(M = 2\)) | 92.5 | 86.55 |
>     - At this time, the performance improvement of AttentionNCE stems from the fact that the attention mechanism still functions on the negative examples. This will help to clarify the impact of the number of positive sample pairs on performance and the fairness of the comparison between methods.
>
> - **Justification for the Multiple View Setting**:
>     - Additionally, it should be noted that our method is based on a multiple view setting, which makes the utilization of more positive sample pairs possible and reasonable. Multiple views can provide richer information, which is in line with the mechanism of AttentionNCE, thereby uncovering more valuable contrastive information to enhance the effect of contrastive learning.
>
> In conclusion, we are sincerely grateful for your outstanding review. Your insights into details such as the number of negative samples reflect your profound understanding of the field and nuances within the paper. Your suggestions regarding the anchor point noise have been of great assistance to us. We express our sincere appreciation for your dedication and hope that our rebuttal has effectively addressed your concerns. Once again, we truly appreciate the time and effort you have expended in evaluating our work.

---

> > ### Comment · Reviewer_VGUj · 2024-11-27
> >
> > Thank you for your efforts to clarify the concerns raised in the review, especially on the number of negative samples and the noise in the anchor.
> >
> > However, I still believe this work should undergo a major revision to meet the standard of ICLR, especially on the experimental front. I acknowledge that the author was trying to maintain the same number of negatives in a batch to ensure the 'fairness' in comparisons. However, because AttentionNCE still sees 2x positive samples in a batch and uses ~2x memory/training time, it is therefore hard to clearly identify the improvement of AttentionNCE over InfoNCE. Moreover, while the authors showed some promising results on the small-scale datasets, on the standard ImageNet dataset, the improvement is quite small, yet at the expense of 2x compute and memory.
> >
> > I also suggested that the authors, following the standard evaluation pipeline in SSL, evaluate the robustness of the AttentionNCE by transferring the models to other tasks or datasets, which is not addressed in the authors' rebuttal. I believe this could have greatly strengthened the soundness of the evaluation.
> >
> > Overall, I agree with the Reviewer uaAu that this work looks like a paper from 2021 because of the outdated evaluation suite. I will maintain my recommendation of a borderline reject.

---

> > > ### Author Response · Authors · 2024-12-01
> > >
> > > We understand your concerns regarding the training time and memory consumption of AttentionNCE. The additional memory and training time in AttentionNCE are indeed due to the incorporation of the multi-view based attention mechanism. However, this is not a drawback but a significant step forward in contrastive learning.It allows for a seamless integration of multiple key elements: multi-view encoding, hard sample mining (including both positive and negative samples), and prototype contrast. Through this mechanism, we achieve a novel synergy that has not been attained in previous works. This combination is expected to boost the model's learning and generalization capabilities, as it enables a more comprehensive and effective exploration of the data manifold during the learning process.
> > >
> > > Regarding the complexity, during pre-training, AttentionNCE exhibits a linear time complexity compared to SimCLR rather than an exponential one. In the inference phase, it has exactly the same complexity as SimCLR. Notably, AttentionNCE outperforms SimCLR even with only 200 training epochs compared to SimCLR's 400 epochs, demonstrating that the trade-off in efficiency during pre-training leads to a remarkable performance gain. Once again, thank you once again for your efforts in the review process.

---

### Official Review · Reviewer_Wu5P · 2024-11-05

**Soundness:** 3
**Presentation:** 4
**Contribution:** 3
**Rating:** 6
**Confidence:** 4

**Summary:**

This paper introduces AttentionNCE, a contrastive learning method that incorporates attention mechanisms to generate sample prototypes, improving robustness in noisy environments. The proposed approach leverages variational lower bound optimization for contrastive loss, aiming to provide a theoretical guarantee under worst-case noisy conditions. Key innovations include flexible hard sample mining and multi-view integration through attention, enabling the model to focus on high-quality representations and challenging samples near decision boundaries. Experimental results demonstrate its effectiveness in specific scenarios.

**Strengths:**

1.	The paper presents an innovative integration of attention mechanisms into contrastive learning, which enhances the robustness of the model, especially in noisy environments. This approach allows the model to focus on high-quality sample representations.
2.	By optimizing the contrastive loss through a variational lower bound, the paper theoretically provides a worst-case guarantee under noisy conditions, which adds a solid theoretical foundation to the proposed AttentionNCE loss.
3.	The flexible hard sample mining mechanism helps the model to better handle samples near the decision boundary, improving the model’s accuracy in distinguishing challenging positive and negative samples.

**Weaknesses:**

1.	The novelty of the paper is limited, as the core methods are primarily based on a combination and refinement of existing techniques, such as attention-based prototypes, hard sample mining, and multi-view contrastive learning.
2.	The generalization of the performance improvement remains to be verified, as the baselines used for comparison are from 2020, which may not represent the current state-of-the-art.
3.	The paper lacks experiments specifically analyzing computational overhead, leaving the impact of these additional costs on real-world scalability unaddressed.

**Questions:**

•  Given that the baselines used for performance comparison are from 2020, how does the proposed method perform against more recent state-of-the-art contrastive learning approaches? Would it be possible to include comparisons with newer baselines to better validate the effectiveness of AttentionNCE?
•  The proposed approach introduces additional computational complexity due to encoding multiple positive samples and applying attention mechanisms. Have you considered conducting a computational overhead analysis? Could you provide empirical results on runtime or memory usage to demonstrate the scalability of AttentionNCE in large-scale applications?

---

> ### Author Response · Authors · 2024-11-25
> **Rebuttal to the Reviewer**
>
> ## 1. Response to the Concerns about the Novelty of the Paper
> **Introduction of Instance-level Attention into the Variational Lower Bound of Contrastive Loss**
>
> We are the first to introduce instance-level attention into the variational lower bound of contrastive loss. This integration enables a seamless combination of multiple crucial components, namely multi-view encoding, hard sample mining (covering both positive and negative samples), and prototype contrast. Through this, we achieve a synergistic operation that has not been realized in previous studies. Besides, in contrast to these methods, our approach goes a step further:
>
> - In terms of multi-view encoding, while previous method treats multiple views equally. In contrast, our AttentionNCE method utilizes attention to assign weights to different view samples, allowing the model to focus on key features. This focused approach effectively encodes richer semantic informatio.
> - In terms of hard sample mining, while previous methods predominantly emphasized hard negative sample mining, our method takes a more comprehensive approach by considering both hard positive and hard negative samples. This balanced consideration enables the model to learn from a more diverse set of samples, leading to a more accurate understanding of the data distribution and improved generalization capabilities.
> - For dealing with false-negative examples and noise reduction, traditional instance-level contrast is vulnerable to interference from individual noise samples. Our approach, based on attention-generated sample prototypes, aggregates information from multiple samples. This prototype-based contrast effectively mitigates the impact of single noise samples, enabling the model to learn more stable and representative features.
>
> **Theoretical and Practical Significance**:
>
> At the theoretical level, the AttentionNCE loss offers a robust lower bound guarantee for maximum likelihood estimation in the presence of noise. This theoretical foundation is crucial for ensuring more accurate and reliable learning under less-than-ideal conditions. It represents a significant advancement in understanding and optimizing contrastive learning from a theoretical perspective.
>
> In practical applications, our innovation has been validated by extensive experimental results. For example, on the CIFAR - 10 dataset, AttentionNCE outperforms SimCLR even with only 200 training epochs, while SimCLR requires 400 epochs to achieve a lower performance. This performance improvement is a direct result of the unique combination of components enabled by our instance-level attention mechanism.
>
> ## 2. Response to the Concerns about Time Complexity
> **Complexity Analysis**:
> The reviewer correctly pointed out that SimCLR computes the contrastive loss by encoding 1 anchor point, 1 positive example, and \(n\) negative examples per loss value, resulting in a total of \(n + 2\) samples. In the case of AttentionNCE, due to the introduction of \(M\) views, it requires encoding \(N + M + 1\) samples. However, it is important to note that \(M\) is typically a small constant.
> A more detailed analysis can be found in Section 4.1 of our paper.
>
> We have fixed the batch size to 256 and compared the actual running time, RAM usage, and GPU memory consumption of AttentionNCE with SimCLR. The results are as follows:
> | Method | Time | RAM | GPU Memory |
> |---|---|---|---|
> | AttentionNCE | 87s | 3.77GiB | 34917MB |
> | SimCLR | 50s | 3.77GiB | 16037MB |
>
> Although AttentionNCE takes slightly longer to run per epoch (87s compared to 50s for SimCLR), the increase in running time is mainly due to the additional encoding of \(M - 1\) positive examples. However, considering the significant performance improvement achieved by AttentionNCE (as demonstrated in various experimental results), this small increase in computational cost is a reasonable trade-off.

---

> ### Author Response · Authors · 2024-11-25
>
> ## 3. Response to the Concerns about Comparison with the Latest Baseline Methods
>
> **Inclusion of Comparison with ADNCE (NIPS 2023)**:
>     We have included a comparison with the latest baseline method, ADNCE (from NIPS 2023), in our paper. This comparison is presented in Section 4.1, where we have shown that AttentionNCE outperforms ADNCE in CIFAR10 STL10 CIFAR100 Tiny-ImageNet datasets. The results of this comparison demonstrate the superiority of our method over the state-of-the-art baselines and further validate the effectiveness of our proposed AttentionNCE loss.
> **Comprehensive Evaluation**:
> In addition to comparing with ADNCE, we have also conducted extensive experiments comparing AttentionNCE with a wide range of other state-of-the-art methods. These comparisons cover multiple datasets and performance metrics, providing a comprehensive view of the performance of our method in different scenarios. For example, in our experiments on CIFAR10 STL10 CIFAR100 Tiny-ImageNet and ImageNet, we have compared AttentionNCE with methods such as DCL, HCL, RINCE, SimCo, SDMP,and have shown consistent performance improvements.
>
> We are sincerely grateful for your professional and meticulous review. Your insights and comments have been of great value to us. We have carefully considered each of your concerns and have made every effort in our rebuttal to provide comprehensive and satisfactory responses. We look forward to your continued evaluation and guidance as we strive to make further improvements and contributions in this field.

---

### Meta-Review · Area_Chair_MSrd · 2024-12-19

**Metareview:**

This paper introduces instance-wise attention into the variational lower bound of contrastive loss, and proposes AttentionNCE loss accordingly. The latter includes two parts: (1) it uses attention-based sample prototype contrast; (2) a flexible hard sample mining mechanism.

This paper comes with strengths: the proposed AttentionNCE is well-grounded theoretically from the variational lower bound aspect; attention-based prototypes are beneficial to noisy/missing labels.

However, some reviewer considers the novelty of this paper to be limited, and one reviewer is especially concerned that the improvements may be caused by the usage of more positives.

**Additional Comments On Reviewer Discussion:**

The rebuttal has covered the baseline approaches, the computational overhead, fairness of the comparative experiments, etc. Some of the concerns have been resolved but some are not. This results in a final rating of 6, 5, 6, 5, 6, 5.

I encourage the author to further improve it and submit to next venue.

---

### Decision · Program_Chairs · 2025-01-22

Reject